# Preferences of people living with HIV for differentiated care models in Kenya: A discrete choice experiment

Sagar Dommaraju[1]*, Jill Hagey[2], Thomas A. Odeny[3,4], Sharon Okaka[4], Julie Kadima[4], Elizabeth A. Bukusi[4], Craig R. Cohen[1,5], Zachary Kwena[4], Ingrid Eshun-Wilson[5], Elvin Geng[5]

1 Department of Global Health, University of California San Francisco, San Francisco, California, United States of America, 2 Department of Obstetrics and Gynecology, Duke University Hospital, Durham, North Carolina, United States of America, 3 Department of Medicine, University of Missouri-Kansas City, Kansas City, Missouri, United States of America, 4 Kenya Medical Research Institute, Nairobi, Kenya, 5 Department of Medicine, University of California San Francisco, San Francisco, California, United States of America

* Sagar.Dommaraju@northwestern.edu

## Abstract

### Introduction

To improve retention on HIV treatment in Africa, public health programs are promoting a family of innovations to service delivery—referred to as "differentiated service delivery" (DSD) models—which seek to better meet the needs of both systems and patients by reducing unnecessary encounters, expanding access, and incorporating peers and patients in patient care. Data on the relative desirability of different models to target populations, which is currently sparse, can help guide prioritization of specific models during scale-up.

### Methods

We conducted a discrete choice experiment to assess patient preferences for various characteristics of treatment services. Clinically stable people living with HIV were recruited from an HIV clinic in Kisumu, Kenya. We selected seven attributes of DSD models drawn from literature review and previous qualitative work. We created a balanced and orthogonal design to identify main term effects. A total of ten choice tasks were solicited per respondent. We calculated relative utility (RU) for each attribute level, a numerical representation of the strength of patient preference. Data were analyzed using a Hierarchical Bayesian model via Sawtooth Software.

### Results

One hundred and four respondents (37.5% men, 41.1 years mean age) preferred receiving care at a health facility, compared with home-delivery or a community meeting point (RU = 69.3, -16.2, and -53.1, respectively; p << 0.05); receiving those services from clinicians and pharmacists—as opposed to lay health workers or peers (RU = 21.5, 5.9, -24.5; p < 0.05); and preferred an individual support system over a group support system (RU = 15.0 and 4.2; p < 0.05). Likewise, patients strongly preferred longer intervals between both clinical

**Data Availability Statement:** All relevant data are within the paper and its Supporting Information files. We have included the raw data as a .db3 file, which can be accessed by software that uses the

SQL programming language. The data was stored this way by the software that we used for our study.

**Funding:** EG received a grant from the National Institutes of Health under the terms of K24 AI134413 (https://www.nih.gov/). SD received a travel grant from the University of California-San Francisco to cover international transportation and housing costs for the primary author. The funders had no role in study design, data collection and analysis, decision to publish, or preparation of the manuscript.

**Competing interests:** The authors have declared that no competing interests exist.

reviews (RU = 40.1 and -50.7 for 6- and 1-month spacing, respectively; p < 0.05) and between ART collections (RU = 33.6 and -49.5 for 6- and1-month spacing, respectively; p < 0.05).

## Conclusion

Although health systems find community- and peer-based DSD models attractive, clinically stable patients expressed a preference for facility-based care as long as clinical visits were extended to biannual. These data suggest that multi-month scripting and fast-track models best align with patient preferences, an insight which can help prioritize use of different DSD models in the region.

## Introduction

Despite greatly expanded access to antiretroviral therapy (ART), about 30% of people living with HIV (PLHIV) in Kenya have not achieved consistently suppressed HIV viral load [1–4]. Treatment for HIV in the country over the last decade has been conducted using a "one-size-fits-all model" that was developed during the emergency response phase of the initial HIV epidemic and which focused on treating patients with advanced medical disease [5]. Patients—regardless of their age, socioeconomic status, CD4 levels, and medical stability—are required to visit the clinic every 1–3 months and to collect their medication just as often [5].

DSD models, which vary the timing, frequency, location and cadre of delivering care, have recently arisen as an alternative to the traditional model and have been heralded as a strategic solution to enhance the effectiveness and efficiency of the care cascade [6–9]. Many patients who are clinically stable need not expend considerable opportunity costs to visit facilities monthly or even quarterly to receive high quality care. Reduction of unnecessary visits would also ameliorate overburdened clinic facilities, reduce waiting time, and potentially improve health care worker burnout. DSD models like multi-month scripting, fast track, adherence clubs, and community ART groups all attempt to reduce visits, offset labor from health care workers to peers and patients, and encourage community-based treatment [5].

As DSD models continue to expand and require maintenance, public health implementers may need to select between models, as implementing multiple models may be burdensome. One way to assign prioritization could be to identify which features are most desired by patients, information which is to date relatively poorly understood. There are many variations of innovations in DSD, and identifying features that are preferred can suggest which are most likely to lead to sustained engagement and retention. For example, DSD includes strategies to deliver HIV care outside of the health facility and deliver care with peers or non-clinicians. Do patients prefer community-based care over facility-based care, and by how much? If they come to the facility rarely, do they prefer higher-cadre health workers, and by how much? DSD models are a large and diverse set of practices, and the comparative desirability of each model may influence which to prioritize. While some data exist on comparative effectiveness of different models, these experimental trials have not accounted for patient preferences which could be an important driver of success for a given individual [10, 11].

We propose to advance our understanding of patient preferences for innovations in HIV care through use of a discrete choice experiment. This discrete choice experiment (DCE) asked patients to choose between hypothetical models of care, which differ in certain attributes such as social support, frequency of clinical visits, and location of visits. In doing so, DCEs

offer an efficient way to examine the potential effectiveness of DSD by quantifying preferences in a population [12, 13]. Services that are most preferred by patients are most likely to be taken up and are most likely to yield consistent engagement. We aimed to reveal which attributes of treatment programs would be most important to patients, which can yield information relevant to policy makers.

## Materials and methods

This study took place in a dedicated HIV clinic in Kisumu County in Western Kenya between April 25, 2017 and June 6, 2017. This HIV clinic is a public health facility operated by the Kisumu County Health Department and supported by Family AIDS Care and Education Services (FACES), a collaboration between the University of California, San Francisco and the Kenya Medical Research Institute [14]. Stable HIV patients were recruited from the facility based on the following criteria: Age greater than or equal to 20 years, BMI greater than 18.5, most recent viral load less than 5,000 copies/mL, on current ART regimen for greater than 12 months, adherent to scheduled clinic visits for the past 6 months, non-pregnant/not breastfeeding, and the primary healthcare team does not have concerns about providing longer follow-up intervals for the patient. These criteria mirror those utilized by the Kenyan Ministry of Health as patients potentially eligible for DSD models [15]. Additionally, the Kenyan Ministry of Health has recently encouraged all patients living with HIV to undergo 6 months of isoniazid prevention therapy; however, this has not been fully implemented at the current HIV clinic and was not included in our inclusion criteria.

Attributes for the discrete choice experiment were identified primarily through a literature search on differentiated care in the region [16–19]. Although not all of these studies were conducted in Kenya, the attributes identified were applicable to HIV care in the country. We confirmed the validity of these attributes by conducting qualitative interviews with the FACES differentiated care team, clinicians at the HIV clinic, and other healthcare personnel who worked with the HIV+ population. Of the original seven attributes that were identified, all seven were ultimately included in our final design. We categorized the levels of each attribute with a similar process. All attributes and their levels are listed below (Table 1).

Questionnaires were produced by using the Choice-based Conjoint feature of Sawtooth Software™, which is widely used for DCE design, administration, and analysis [20]. Choice tasks were generated that maximize balance—meaning each level appears with the same frequency—and orthogonality—meaning each pair of levels appears with the same frequency across each pair of attributes. The experimental design generated 10 choice tasks that were

**Table 1. Attributes and levels of differentiated care models for HIV in Kenya.**

| Attributes | Levels | | | |
|---|---|---|---|---|
| Location of ART refills | Health facility[1] | Community meeting point | Home | |
| Frequency of receiving ART refills | Every month | Every 3 months | Every 6 months | |
| Person providing ART refills | Nurse | Lay health worker | Pharmacist | PLHIV[2] |
| Adherence (adh) support provided[3] | No adh support | Individual adh support | Group adh support | |
| Refill pick-up/delivery times | Weekday during facility hours | Weekday (early morning, evening) | Weekend | |
| Location of viral load sample | Health facility[1] | Community meeting point | Home | |
| Frequency of clinical visits | Every month | Every 3 months | Every 6 months | Every 12 months |

[1]Either HIV-specific or integrated primary care clinic

[2]Either participant or other member of the group, community peer

[3]Individual or group counseling around ART adherence, either by peers or health workers

**Table 2. Example of choice task.**

| If these differentiated care models (DCMs) were your only options, which one would you choose? Please put an X under that DCM. | | | |
|---|---|---|---|
| **Attributes** | **DCM 1** | **DCM 2** | **DCM 3** |
| **Location of ART refills** | **Community meeting point** | **Home** | **Health Center** |
| Frequency of receiving ART refills | every 3 months | Every 3 months | Every month |
| Adherence support provided | No support | Individual support | Group support |
| Person providing ART refills | Pharmacist | PLHIV | Nurse |
| | | X | |

added to the questionnaire, with each choice task asking a patient to choose between three hypothetical care models that have different levels of attributes (Table 2). A forced-choice format (i.e. no option to opt out of all three care models altogether) was used in order to more closely approximate the choice that a patient would have to make in real life, assuming that a stable patient with HIV would necessarily choose any care model over none at all.

Sawtooth generates the design by sampling from a subset of the full-choice designs for each respondent while ensuring level balance and near-orthogonality within each respondent's profile. This allows for the generation of up to 999 blocks, and using a unique randomized design for each respondent reduces context effects. We used a fractional factorial design to reduce the number of choice tasks required in the experiment and removed combinations of attributes and levels that would not be feasible or practical (e.g. patient receiving ART refill at clinic but having viral load taken at home). Finally, we used a partial profile design wherein each choice task was limited to four attributes rather than the total seven. Compared to full profile, a partial profile design reduces the cognitive burden on patients and thus lowers response error, producing results with greater predictive validity [21].

Patients were randomly allocated to receive one of twelve different, randomly-generated versions of the questionnaire, which had been translated to Dholuo and Kiswahili and uploaded onto Sawtooth servers. These questionnaires were accessed via Android tablets at the HIV clinic. Patients were given verbal instructions at the start of the interview and before each choice task to ensure understanding of the care models being presented. Each attribute and level was explained, and patients were allowed to complete the survey at their own pace.

To have enough power to reveal main effects, we needed a minimum of 67 patients based on the following sample size calculation for conjoint-based analysis in discrete choice experiments:

$$\frac{nta}{c} \geq 500$$

where n is the number of patients, t is the number of choice tasks per questionnaire (t = 10 in our study), a is the number of options per choice task (a = 3), and c is the number of cells (c = 4) [22, 23]. To have enough power to examine all two-way interactions between attributes, we needed 267 patients based on the same formula as above, setting c = 16 to reflect the largest product of levels of any two attributes.

Patients were introduced to the study during their clinician visit if they met criteria as a stable patient. One of three researchers approached each interested patient to obtain oral consent, and administered the questionnaire containing sociodemographic information and the ten choice tasks in the language of the patient's choosing. Patient IDs were also collected to obtain additional data from the EMR. Detailed field notes and observations were taken in tandem. Basic demographic information was collected from all participants: age, gender, education level, average monthly income, and average travel time to the clinic.

We used parametric and non-parametric tests to summarize all sociodemographic information using R version 3.2.3 [24]. Significance testing for continuous variables was completed using Student's t-test or Wilcoxon's Rank Sum test where appropriate and for categorical variables using Chi Square or Fisher's exact test where appropriate. Relative utilities and average importance for each of the levels and attributes, respectively, were calculated by Sawtooth Software using Hierarchical Bayesian (HB) analysis and effects coding [22, 24]. Compared to mixed logit or latent class models which may be used to analyze DCE data, HB analysis uses a two-level hierarchical model to generate both relative utilities for the population as well as individual utilities which can be used to identify detailed preference segments in the population. The HB model in Sawtooth has two levels: At the upper level, it is assumed that individuals' vectors of part-worths are drawn from a multivariate normal distribution [25]. At the lower level, a logit model is assumed for each individual, where the utility of each alternative is the sum of the part-worths of its attribute levels, and the respondent's probability of choosing each alternative is equal to its utility divided by the sum of utilities for the alternatives in that choice set [25]. Several Markov Chain Monte Carlo (MCMC) simulations of an algorithm using these model estimates generates the part-worths for the individuals, the mean for the population, and variances and covariances [25]. Represented as a formula, the utility function we used is described by Rao et al as follows:

$$U_t(x_{jt}) = D_{t1}U_{t1} + D_{t2}U_{t2} + \cdots + D_{tr_t}U_{tr_t}$$

$$Ut_{(xjt)} \sim N(Dt_iUt_i)$$

Where $U_t$ = part-worth function of the $t^{th}$ attribute; $x_{jt}$ = level for the $j^{th}$ profile on the $t^{th}$ attribute; $r_t$ = number of levels for the $t^{th}$ attribute; $D_{tk}$ = 1 if the value $x_{jt}$ is equivalent to the $k^{th}$ discrete level of $x_t$ and 0 otherwise; and $U_{tk}$ = component of the part-worth function for the $k^{th}$ discrete level of $x_t$ [26]. The effect of each attribute of our DCE was modeled by:

$$U_t\left(x_{jt}\right) = D_i\left(\frac{t_1 + t_2 + \cdots + t_n}{k}\right) * U_i$$

Where $D_i$ = the discrete value assigned for each of $i^{th}$ attributes; k = the discrete level of $x_t$; and $t_1 + t_2 + \ldots + t_n$ is the sum of the attributes at the $n^{th}$ level.

In addition, average importances are calculated in Sawtooth via standard probability analysis by dividing the utility range for each attribute (i.e. the utility of the highest level minus the utility of the lowest level) by the total sum of utility ranges for all attributes [22]. Average importances are reported as percentages and can be interpreted as how important each attribute is for a patient when making a decision regarding their preferred DSD model [20]. Descriptive statistics were used to describe any differences in relative utilities or importances across various socio demographic groups, although the study was not powered to confidently detect differences between sub-groups. Qualitative data from field notes were summarized using narrative analysis, whereby researchers identified common unifying stories that arose in response to probing questions about patients' decision-making process [27].

This evaluation was approved by the Kenya Medical Research Institute Ethical Review Committee and the University of California San Francisco Human Research Protection Program as part of routine program evaluation within the Family AIDS Care and Education Services (FACES) program.

# Results

## Sociodemographic characteristics

A total of 104 stable patients with HIV were recruited from the HIV clinic in Kisumu over a 5-week period beginning May 2017. The mean age of our study sample was 41 years, and 62.5% of the participants were women (Table 3). Women were significantly younger with a lower income than their male counterparts (Table 4).

## Relative utilities and importances

Relative utilities (RUs) represent patient preference and were calculated for each level of the discrete choice experiment (Table 5).

For location of ART refills, patients strongly preferred a model of care where they received ART at the health center, followed by a home-delivery model, followed by a community-point-delivery model (RU = 49.95, -3.88, and -46.07). The same trends were found for location of clinical review (RU = 69.28, -16.19, and -53.10).

**Table 3. Sociodemographic characteristics of HIV-infected patients (n = 104).**

| Characteristic | Number (%) | Mean | Std Dev |
|---|---|---|---|
| Gender | | | |
| Men | 39 (37.5) | | |
| Women | 65 (62.5) | | |
| Other | 0 (0.0) | | |
| Age (yrs) | | 41.06 | 10.89 |
| 18–34.9 | 32 (30.8) | | |
| 35–49.9 | 47 (45.2) | | |
| 50–64.9 | 23 (22.1) | | |
| > = 65 | 2 (1.9) | | |
| Survey Language | | | |
| English | 51 (49.0) | | |
| Kiswahili | 12 (11.5) | | |
| Dholuo | 41 (39.4) | | |
| Education Level[1] | | | |
| None | 1 (1.0) | | |
| Primary | 50 (48.1) | | |
| Secondary | 41 (39.4) | | |
| Univ/Postgrad | 12 (11.5) | | |
| Income (KSH/mo) | | 9161 | 12966 |
| Below PL[2] | 61 (58.7) | | |
| Low income[3] | 36 (34.6) | | |
| Middle income+ | 7 (6.7) | | |
| Travel time (min) | | 48.08 | 44.9 |
| 0–29.9 | 26 (25.0) | | |
| 30–59.9 | 47 (45.2) | | |
| > = 60 | 31 (29.8) | | |

[1]Indicates highest level of education started or completed

[2]PL = Poverty line, monthly income below KSH 6200

[3]Low income = Monthly income between KSH 6200 and 26000

**Table 4. Comparison of continuous and categorical variables by gender amongst HIV-infected patients (n = 104).**

| | Gender | | | |
|---|---|---|---|---|
| Continuous Variable | Men | Women | test stat | p |
| Age, mean (yrs) | 44.64 | 38.91 | t = 2.553 | *0.013 |
| Income, median (KSH/mo) | 7000 | 4000 | W = 1590 | *0.03 |
| Travel time, median (min) | 30 | 30 | W = 1373 | 0.466 |
| Categorical Variable | Men | Women | test stat | p |
| Survey Language | | | $\chi^2 = 5.92$ | 0.052 |
| English, n | 22 | 29 | | |
| Kiswahili, n | 7 | 5 | | |
| Dholuo, n | 10 | 31 | | |
| Education Level | | | Fisher | 0.858 |
| None, n | 0 | 1 | | |
| Primary, n | 17 | 33 | | |
| Secondary, n | 17 | 24 | | |
| Univ/Postgrad, n | 5 | 7 | | |

* significant at p < 0.05.

** significant at p < 0.01.

Histogram of all significant differences provided on the right.

For frequency of ART refills, patients preferred a model of care where they only had to collect their drugs every 6 months, followed closely by a model with ART refills every 3 months (RU = 33.63 and 15.89). They overwhelmingly did not prefer a 1-month ART refill system

**Table 5. Average utilities of all levels included in our study.**

| Attributes and Levels | Avg Utilities | 95% CI | Std Dev | 95% CI | Attributes and Levels | Avg Utilities | 95% CI | Std Dev | 95% CI |
|---|---|---|---|---|---|---|---|---|---|
| Location of ART Refills | | | | | Adherence Support | | | | |
| Health Centre | 50.0 | (37.45, 62.45) | 64.26 | (,) | No support | -19.3 | (-23.77, -14.76) | 23.15 | (,) |
| Community meeting point | -46.1 | (-56.00, -36.14) | 51.05 | (,) | Individual support | 15.0 | (10.71, 19.35) | 22.19 | (,) |
| Home | -3.9 | (-14.41, 6.65) | 54.15 | (,) | Group support | 4.2 | (-0.61, 9.08) | 24.92 | (,) |
| Location of Clinical Review | | | | | Person delivering ART | | | | |
| Health Centre | 69.3 | (57.95, 80.62) | 58.30 | (,) | Nurse | -2.8 | (-6.88, 1.22) | 20.81 | (,) |
| Community meeting point | -53.1 | (-61.83, -44.36) | 44.93 | (,) | Pharmacist | 21.4 | (17.09, 25.80) | 22.39 | (,) |
| Home | -16.2 | (-27.80, -4.58) | 59.71 | (,) | PLHIV (community peer) | -24.5 | (-29.04, -19.94) | 23.40 | (,) |
| Frequency of ART Refills | | | | | Lay health worker | 5.9 | (1.35, 10.40) | 23.27 | (,) |
| Every month | -49.5 | (-56.38, -42.67) | 35.24 | (,) | Refill delivery time | | | | |
| Every 3 months | 15.9 | (12.08, 19.71) | 19.60 | (,) | Weekday (regular hours) | 16.3 | (11.61, 21.02) | 24.18 | (,) |
| Every 6 months | 33.6 | (27.80, 39.46) | 29.96 | (,) | Weekday (off hours) | -1.8 | (-6.45, 2.85) | 23.90 | (,) |
| Frequency of Clinical Rev | | | | | Weekend | -14.5 | (-18.58, -10.44) | 20.93 | (,) |
| Every month | -50.7 | (-56.85, -44.56) | 31.58 | (,) | | | | | |
| Every 3 months | 7.4 | (3.80, 10.96) | 18.42 | (,) | | | | | |
| Every 6 months | 40.1 | (33.52, 46.72) | 33.95 | (,) | | | | | |
| Every 12 months | 3.2 | (-2.02, 8.44) | 26.90 | (,) | | | | | |

A higher magnitude (or darker shade) indicates a stronger preference, while a positive or negative value (green or red) indicates a positive or negative preference.

*Red represents the least preferred attribute level and green represents the most preferred.

*All differences are statistically significant at p < 0.05, except for 3 months vs 12 months for Frequency of Clinical Review

relative to the other two options (RU = -49.53). Similarly, for frequency of clinical review, patients strongly preferred a model of care where they had a 6-month time between clinical appointments (TCA), followed by either a 3-month or 12-month TCA, and a strong preference against a 1-month TCA (RU = 40.12, 7.38, 3.21, and -50.71).

For adherence support provided, patients preferred an individual support system (i.e. one-on-one counseling) over a group support system. However, they preferred either option over having no support at all (RU = 15.03, 4.23, and -19.27). Patients preferred that the person delivering ART be a pharmacist, followed by a lay health worker, followed by a nurse, and finally by a community peer/PLHIV (RU = 21.45, 5.87, -2.83, and -24.49). Finally, for refill pick-up/delivery times, patients preferred to collect their drugs during regular clinic hours—between 8 AM and 4 PM—instead of off-hours or on the weekend (RU = 16.31, -1.80, and -14.51).

The average importance of each attribute of a care model is presented in Table 6. The location of clinical review and the location of ART refills were the most important attributes in the decision-making process when patients were faced with the choice tasks, and both were significantly more important to patients than any other attributes (p < 0.05).

The next most important attributes were the frequency of clinical review and the frequency of ART refill, with average importances of 15.18 and 13.50, respectively. These are followed in importance by the person providing ART, the form of adherence support provided, and the schedule for ART refill delivery (Avg. Importance = 9.65, 8.19, and 7.78, respectively).

## Differences in DCE results by gender

Differences in relative utilities between men and women were also assessed (Table 7). Both men and women strongly preferred to receive their ART at a health center than at a community meeting point or at home, but women found home-based delivery more acceptable than did men (RU = 9.76 and -24.47, respectively; p < 0.05). Both genders also preferred to receive their drugs from a pharmacist rather than a nurse, lay health worker, or peer. Women, however, found it more acceptable to receive ART refills from lay health workers than did men (RU = 20.80 and -4.59; p < 0.05).

Both men and women strongly disliked receiving their ART refills on a monthly basis, preferring instead to receive them at three, six, or twelve-month intervals. Men, however, had almost equal preference for three and six-monthly refills (RU = 28.1 and 36.0), whereas women highly preferred the latter (RU = 5.2 and 47.4; p < 0.05). Both genders preferred six-monthly clinical review compared to one, three, or twelve-monthly options. Finally, men reported a stronger preference for a group-based support system than did women (RU = 25.01

**Table 6. Average importances of all attributes included in our study.**

| Attribute | Avg Importances (%) | Std Dev | 95% CI |
|---|---|---|---|
| Location of clinical review | 24.1 | 6.6 | (22.8, 25.4) |
| Location of ART Refills | 21.6 | 7.6 | (20.1, 23.1) |
| Frequency of clinical visits | 15.2 | 6.0 | (14.0, 16.3) |
| Frequency of ART refill | 13.5 | 7.6 | (12.0, 15.0) |
| Person providing ART | 9.7 | 3.9 | (8.9, 10.4) |
| Adherence Support | 8.2 | 4.1 | (7.4, 9.0) |
| Refill pick-up/deliv time | 7.8 | 3.9 | (7.0, 8.5) |

The values represent what proportion of a patient's decision was made based on that attribute. A higher value indicates that the attribute is considered more important by patients when choosing a model of care.

**Table 7. Normalized average utilities of all levels by gender.**

| Attributes and Levels | Gender | | | | | | | |
|---|---|---|---|---|---|---|---|---|
| | Men | | | | Women | | | |
| | Avg Utilities | 95% CI | Std Dev | 95% CI | Avg Utilities | 95% CI | Std Dev | 95% CI |
| Location of ART Refills | | | | | | | | |
| Health Centre | 62.2 | (39.47, 84.99) | 70.2 | (61.79, 81.31) | 45.1 | (26.68, 63.56) | 74.4 | (65.49, 86.17) |
| Community meeting point | -37.8 | (-54.28, -21.26) | 50.9 | (44.82, 58.97) | -54.9 | (-69.68, -40.08) | 59.7 | (52.57, 69.17) |
| Home* | -24.5 | (-38.24, -10.70) | 42.5 | (37.38, 49.19) | 9.8 | (-5.38, 24.90) | 61.1 | (53.78, 70.77) |
| Location of Clinical Review | | | | | | | | |
| Health Centre | 57.0 | (37.69, 76.31) | 59.6 | (52.43, 68.99) | 57.0 | (41.09, 73.01) | 58.3 | (51.31, 67.51) |
| Community meeting point | -43.0 | (-54.27, -31.73) | 34.8 | (30.60, 40.26) | -43.0 | (-57.62, -28.29) | 59.3 | (52.19, 68.67) |
| Home | -14.0 | (-29.89, 1.88) | 49.0 | (43.12, 56.74) | -14.1 | (-33.07, 4.87) | 60.3 | (53.07, 69.82) |
| Frequency of ART Refills | | | | | | | | |
| Every month | -64.0 | (-76.34, -51.76) | 37.9 | (33.36, 43.90) | -52.6 | (-62.29, -42.94) | 39.1 | (34.37, 45.23) |
| Every 3 months* | 28.1 | (21.39, 34.80) | 20.7 | (18.20, 23.95) | 5.2 | (-1.46, 11.92) | 27.0 | (23.76, 31.26) |
| Every 6 months | 36.0 | (23.19, 48.71) | 39.4 | (34.65, 45.60) | 47.4 | (40.26, 54.50) | 28.7 | (25.29, 33.28) |
| Frequency of Clinical Rev | | | | | | | | |
| Every month | -62.7 | (-75.57, -49.80) | 39.8 | (34.99, 46.04) | -45.8 | (-52.92, -38.67) | 28.8 | (25.30, 33.29) |
| Every 3 months | 7.0 | (0.21, 13.85) | 21.0 | (18.52, 24.36) | 5.8 | (-0.48, 12.01) | 25.2 | (22.19, 29.19) |
| Every 6 months | 37.3 | (26.18, 48.45) | 34.4 | (30.24, 39.79) | 54.2 | (46.12, 62.29) | 32.6 | (28.71, 37.78) |
| Every 12 months* | 18.3 | (10.05, 26.62) | 25.6 | (22.49, 29.60) | -14.2 | (-21.12, -7.23) | 28.0 | (24.68, 32.47) |
| Adherence Support | | | | | | | | |
| N o support | -62.5 | (-69.84, -55.16) | 22.6 | (19.93, 26.22) | -43.1 | (-49.53, -36.72) | 25.8 | (22.74, 29.92) |
| Individual support | 37.5 | (26.44, 48.55) | 34.1 | (30.02, 39.50) | 56.9 | (52.27, 61.48) | 18.6 | (16.36, 21.52) |
| Group support* | 25.0 | (15.87, 34.15) | 28.2 | (24.81, 32.65) | -13.8 | (-21.38, -6.13) | 30.8 | (27.08, 35.63) |
| Person delivering ART | | | | | | | | |
| Nurse | 3.1 | (-7.18, 13.42) | 31.8 | (27.96, 36.79) | -10.9 | (-15.40, -6.48) | 18.0 | (15.84, 20.85) |
| Pharmacist | 50.7 | (44.07, 57.40) | 20.5 | (18.08, 23.80) | 45.1 | (37.97, 52.18) | 28.7 | (25.23, 33.20) |
| PLHIV (community peer) | -49.3 | (-59.91, -38.62) | 32.9 | (28.91, 38.04) | -54.9 | (-64.27, -45.59) | 37.7 | (33.17, 43.64) |
| Lay health worker* | -4.6 | (-18.16, 8.98) | 41.9 | (36.85, 48.49) | 20.8 | (15.14, 26.46) | 22.9 | (20.11, 26.47) |
| Refill delivery time | | | | | | | | |
| Weekday (regular hours) | 47.4 | (36.57, 58.22) | 33.4 | (29.40, 38.68) | 51.8 | (46.20, 57.38) | 22.5 | (19.85, 26.11) |
| Weekday (off hours) | 5.2 | (-3.49, 13.90) | 26.8 | (23.61, 31.07) | -3.6 | (-10.09, 2.93) | 26.3 | (23.12, 30.42) |
| Weekend | -52.6 | (-60.91, -44.30) | 25.6 | (22.55, 29.67) | -48.2 | (-53.76, -42.66) | 22.4 | (19.72, 25.94) |

A higher magnitude indicates a stronger preference, while a positive or negative value indicates a positive or negative preference. A star (*) denotes a level where preferences were significantly different between men and women (p < 0.05).

and -13.75; p < 0.05). Though both genders still believed individual support was the best option, men viewed group support as a close second, whereas women did not.

## Field notes and observations

Field notes were taken during each interview by the researcher conducting the questionnaire. If patients cited a specific reason for their decision in a choice experiment, this was documented in notes that were reviewed at the end of the study period.

A majority of patients interviewed preferred models of care in which ART delivery and clinical review were both performed at the health center, as opposed to a community-meeting point or at home. Stigma against HIV was most commonly cited as the reason for this preference.

Patients almost unanimously preferred models of care with longer periods of time between either clinical reviews or ART deliveries. Work and travel restrictions were cited as the most common reasons for this preference

Most patients strongly preferred to receive their drugs from the pharmacist because they view pharmacists as those with the most training for the task. However, a few patients had interacted with lay health workers before, and were fine with models of care that utilize these individuals to take charge of ART delivery.

Patients unanimously preferred some form of social support over none at all. However, given the choice between an individual support or a group support system, patients chose the former. Patients reported fear of stigma, lack of individualized attention to understand how to adhere to one's medications, and challenges in managing a group dynamic as reasons they did not want a group support system.

Finally, most patients preferred care models in which ART delivery and clinical review happened during regular clinic hours simply because this is what they were used to. However, a few patients saw value in off-hours care, particularly if it allowed them more flexibility in their work.

## Discussion

Our discrete choice experiment revealed clear preferences of certain features of HIV care and treatment among PLHIV in Kenya with immediate implications for public health programming. Patients desired to come to the facility less frequently, with six-month intervals showing the greatest utilities. Patients, however, also valued location of care highly when choosing between models: counter to expectations, they strongly preferred models where clinical review and ART refills were done at a central location (e.g. the health clinic) instead of at home or in the community. Patients strongly preferred to interact with healthcare professionals: they preferred physicians/pharmacists rather than peers, and they preferred individual psychosocial support rather than group therapy.

Our findings support previous literature that suggests that excessive visits to the facility for clinical review or medication refill represents an undesired barrier. At present, standard of care in Kenya has increased visit intervals from monthly to quarterly [14], but these data suggest even longer intervals would be more preferred. Six months appeared to be the optimal period, as patients felt a decreasing utility associated with yearly visits for clinical review [28]. These data also mirror observation studies that suggest that a visit interval of six months was associated with a smaller chance of a missed visit as compared to shorter assigned return intervals. While at present the Kenya Ministry of Health recommends 3 months as an upper limit for ART to accommodate supply chain and stocking [28, 29], these data suggest a potential public health benefit to developing the capability to procure, maintain, and distribute enough ART to dispense 6-months of medications.

Many studies have shown the efficacy of community-based forms of HIV care in Kenya and SSA at large in reducing the burden on the facility, alleviating clinician workload, and reducing frequency of clinical appointments (TCA) [30–32]. Although community-based care seems beneficial from a systems standpoint, we found that patients themselves strongly prefer facility-based care models, and patients would often choose a facility-based care model even if it meant sacrificing other benefits such as reduced travel time, individual counseling, or less frequent TCA. Additionally, other studies have proposed using peer health workers and other lay health providers to help decrease health facility staff workloads. However, our study indicates that the person responsible for delivering care was also an important attribute for patients. Patients strongly preferred to receive care by facility-based workers (i.e. clinicians

and pharmacists), and not from PLHIV. One predominant barrier to the implementation of community- or peer-based models of care is the continued high level of stigma and discrimination against patients with HIV in Kenya, which was noted multiple times from field notes during the study [33–35].

Participants in our study prefer to have some form of psychosocial support to help them manage their condition. We observe that many clinics in the region have already led the way in the design and implementation of group support systems for PLHIV, albeit with varying levels of success [36]. However, given the option between an individual support system—such as one-on-one counseling—and a group support system, patients showed a preference towards the former.

Kenya's ongoing push for universal healthcare must be considered when examining our study results. Barriers to this goal include the gap in healthcare access between rural and urban communities, redundancies in service delivery, and a relative shortage of healthcare workers [37, 38]. In addition to being the model most preferred by the patients in our study, a multi-month scripting, fast-track model of care would also help improve the efficiency of the healthcare system and reduce the amount of time spent by workers on stable patients. As HIV/AIDS remains amongst the leading causes of death and disability in the country, adaptation of this model would aid in Kenya's efforts to refinance and restructure the healthcare system to provide universal coverage [37, 38].

Our study's primary strength lies in its design. DCEs provide more useful information than does traditional qualitative research in several ways. First, our study allowed us to not only rank, but also to quantify the magnitude of patient preferences for each attribute and corresponding levels. Such information will be useful for implementors who need to prioritize between several service characteristics when designing a DSD model. Second, our questionnaires simulated choices that patients may have to make in real life, which provides more earnest insight into a patient's preferences than would a traditional survey. Finally, we used prohibitions to ensure that all randomly generated care models would be feasible, so that patients would not be choosing between options that could never exist. Some of our findings support previous research, but many also challenge conventional approaches to differentiated care in Kenya.

However, our study had a few limitations. First, we did not meet the sample size goal to examine all two-way interactions between attributes. Our sample was large enough to reveal all significant within-attribute differences, but not large enough to analyze how preferences might vary between certain sociodemographic characteristics like income or level of education. Future research should prioritize understanding how preferences vary amongst different groups of patients. In addition, it will be important to explore how a patient's preference for one aspect of a care model might influence their preference for another (e.g. how does the preference for location of ART delivery interact with preference for type of support system). Second, we recruited from a naïve population of stable patients in Kisumu. None of the patients interviewed had any previous exposure to any form of differentiated care, so all of their responses were based on what they deemed to be the hypothetical best option. However, this may account for some of the differences seen in which differentiated care choices were acceptable in this study versus other studies where patients were assessed for their thoughts on a model after participating in it. For example, one year after implementation of a community ART group in Lesotho, patients reported overall satisfaction with the care model, citing reduced stigma against HIV in their community in addition to the expected benefits of reduced visit time and increased retention [39].

Finally, the COVID-19 pandemic may reshape patient preferences and health systems practices in a way that we cannot predict with our current data. Certain changes to HIV care in

Kenya in response to COVID-19 coincide with findings from our study. For example, the Kenyan Ministry of Health was recently able to procure and distribute an additional 3-month supply of ART to all FACES HIV clinics in Kenya, effectively increasing the space between ART refills leading to a 50.7% reduction in average daily clinic attendance [40]. This practice coincides with our finding that patients strongly prefer longer spacing between ART refills. However, other changes may be inconsistent with findings from our study. For example, FACES and the Kenyan Ministry of Health have created a goal to scale-up community distribution of ART in order to further decongest clinics and thus reduce transmission of COVID-19 [36]. It is conceivable that patient preference may shift to community- and home-delivery of ART in light of COVID-19. While we cannot assess this and other potential changes with our current data, we believe that this DCE will be useful to ensure that health systems provide the right mix of choices to meet changing patient preferences.

## Conclusions

Differentiated care promises to reduce system inefficiencies—such as unnecessary resources used on stable patients—while simultaneously improving patients' experiences with and retention in treatment regimens. As new research emerges and clinics throughout the country begin to test early forms of differentiated care, special attention must be paid to consider patient preferences in the design and implementation of these care models. Most importantly, we found that patients strongly prefer to stay in a centralized model of HIV care—one in which care is delivered in a health facility by trained health workers. This data must be taken into account in conjunction with the realities of limitations in care provision, but we must remain diligent in finding ways to improve HIV care in Kenya without sacrificing the needs and desires of patients themselves.

## Supporting information

**S1 Dataset. Raw data generated by Sawtooth Software based on the questionnaires submitted by the patients in our study.** This data can only be accessed by software that uses SQL programming language, such as Sawtooth.
(XLSX)

## Acknowledgments

Thank you to the directors of the Global Health program at the University of California-San Francisco who supported this research academically. Finally, a special thanks to the clinicians and staff at Lumumba health clinic in Kisumu, Kenya where this research was conducted.

## Author Contributions

**Conceptualization:** Sagar Dommaraju, Thomas A. Odeny, Elvin Geng.

**Data curation:** Sagar Dommaraju, Sharon Okaka, Elvin Geng.

**Formal analysis:** Sagar Dommaraju, Elvin Geng.

**Funding acquisition:** Sagar Dommaraju, Elvin Geng.

**Investigation:** Sagar Dommaraju, Thomas A. Odeny, Elvin Geng.

**Methodology:** Sagar Dommaraju, Jill Hagey, Thomas A. Odeny, Ingrid Eshun-Wilson, Elvin Geng.

**Project administration:** Sagar Dommaraju, Sharon Okaka, Elvin Geng.

**Resources:** Sagar Dommaraju, Thomas A. Odeny, Elvin Geng.

**Software:** Sagar Dommaraju, Elvin Geng.

**Supervision:** Sagar Dommaraju, Thomas A. Odeny, Craig R. Cohen, Elvin Geng.

**Validation:** Sagar Dommaraju, Thomas A. Odeny, Elvin Geng.

**Visualization:** Sagar Dommaraju, Elvin Geng.

**Writing – original draft:** Sagar Dommaraju, Jill Hagey, Thomas A. Odeny, Ingrid Eshun-Wilson, Elvin Geng.

**Writing – review & editing:** Sagar Dommaraju, Jill Hagey, Thomas A. Odeny, Sharon Okaka, Julie Kadima, Elizabeth A. Bukusi, Craig R. Cohen, Zachary Kwena, Ingrid Eshun-Wilson, Elvin Geng.

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
