## [Decision Letter · Decision Letter 0]

15 Sep 2020

PONE-D-20-24745

Preferences of People Living with HIV for Differentiated Care Models in Kenya: A Discrete Choice Experiment

PLOS ONE

Dear Dr. Dommaraju,

Thank you for submitting your manuscript to PLOS ONE. After careful consideration, we feel that it has merit but does not fully meet PLOS ONE’s publication criteria as it currently stands. Therefore, we invite you to submit a revised version of the manuscript that addresses the points raised during the review process.

We look forward to receiving your revised manuscript.

Kind regards,

Matthew Quaife

Academic Editor

PLOS ONE

Journal Requirements:

Reviewers' comments:

Reviewer's Responses to Questions

**Comments to the Author**

1. Is the manuscript technically sound, and do the data support the conclusions?

Reviewer #1: Yes

Reviewer #2: Partly

2. Has the statistical analysis been performed appropriately and rigorously? 

Reviewer #1: Yes

Reviewer #2: No

3. Have the authors made all data underlying the findings in their manuscript fully available?

Reviewer #1: Yes

Reviewer #2: Yes

4. Is the manuscript presented in an intelligible fashion and written in standard English?

Reviewer #1: Yes

Reviewer #2: Yes

5. Review Comments to the Author

Reviewer #1: The authors have addressed the above questions. A couple of comments for the manuscript.

I would like to acknowledge that I have reviewed your paper titled “Preferences of People Living with HIV for Differentiated Care Models in Kenya: A Discrete Choice Experiment.” It is a very important topic as countries aim for patient-centered care as well as adherence to ART among PLHIV.

Below are a few comments that the authors can consider.

1.Although the attribute identification for the DCE is supported by evidence (Literature review and qualitative interviews), the authors do not justify how the final seven attributes and their levels were arrived at. For instance, its not clear how many attributes were identified from the literature review, how many (more or less) attributes were identified from the qualitative interviews and what process was involved in perhaps reducing the number of attributes or having a consensus about the attribute levels. These are important aspects of reporting conjoint analyses (Bridges et al., 2011). Please clarify

2.It is not clear to me why the example choice task, Table 2, has four attributes instead of the seven. Does this mean the “twelve different, randomly generated” questionnaire versions had a different number/set of attributes? (If I interpret this correctly, 12 blocks were generated each with 10 choice tasks – If this is the case then each task, irrespective of the block would have all the seven attributes and three alternatives). On the other hand, the authors might have opted for partial profiles, however, this has not been described anywhere. Please explain.

3.Given that the study took place in one facility, it is surprising that the study questionnaire was not piloted. How can we guarantee that respondents could easily understand the attributes and levels, the number of attributes and alternatives in each task and generally the number of choice tasks per questionnaire did not result in respondent cognitive burden? Please clarify.

4.Table 4, I suggest the units (years, KSH/month and minutes) be bracketed to avoid confusion. For example, the travel time unit could be interpreted as presenting the minimum value (which can be abbreviated as min).

5.The authors state that they did not meet their original sample size goal. This is not true. They actually surpassed their target minimum sample size of 67 respondents, however, this sample size was only optimized to examine main effects but not all possible two-way interactions. To have enabled an examination of these, ‘c’ should have been equal to the largest product of levels of any two attributes (in this case 12). Rephrase this limitation.

Reviewer #2: I have the following comments

1.In the introduction section, provide a brief context of the Kenyan health system and how the current HIV care is delivered. This will help explain why DSD models in Kenya are important. Just a paragraph would do to help any readers to understand the context, understand why a DCE was needed, and understand the policy context. State why the DCE was better than any other method out there.

2.The methodology section is not clear. It omits a lot of crucial information that helps the reader understand what was done.

3.“Patients were introduced to the study during their clinician visit if they met criteria as a stable patient. One of three researchers approached each interested patient to obtain oral consent, and administered the questionnaire containing sociodemographic information and the ten choice tasks in the language of the patient’s choosing. Patient IDs were also collected to obtain additional data from the EMR. Detailed field notes and observations were taken in tandem. Basic demographic information was collected from all participants: age, gender, education level, average monthly income, and average travel time to the clinic.” – This piece of text should be put in the right place which describes data collection. This paragraph should come after sample size calculation because you talk about 10 choice tasks being administered while you haven’t described attributes, levels, experimental design, questionnaire format, and sampling. Therefore, it should be placed further down the methodology section preferably after sampling/sample size calculation.

4.I can see that from Table 2, the DCE was unlabelled with three hypothetical alternatives without an opt-out. You need to state this in the text before the sentence “Questionnaires were produced by using the Choice-based Conjoint feature of Sawtooth Software™. Justify in the text why you opted for a forced-choice format i.e. why was the opt-out not needed.

5.In the construction of choice tasks, did you use partial profiles? Because Table 1 shows 7 attributes while Table 2 shows only 4 attributes. If so, state this in the text why partial profiles were used instead of full profiles.

6.Why was an orthogonal design appropriate for this study while there are far better designs such as efficient and Bayesian efficient designs? State this in the text. Also clearly state that the orthogonal design was a main effects model.

7.The experimental design generated 10 choice tasks which were put into the questionnaire.

8.Were the attributes and levels clearly explained to participants? State this in the text

9.“used parametric and non-parametric tests to summarize all sociodemographic information using R version 3.2.3 [19]”. State these tests?

10.“importances” Relative importance would be a better?

11.“In addition, average importances are calculated and represent the relative importance of each attribute within the experiment. Average importances are presented as percentages and can be interpreted as how important each attribute is for a patient when making a decision regarding their preferred DSD model [17].” Clearly state in the text in the methodology section the method was used to calculate relative importance estimates.

12.Write out the utility functions so we can see the model structure and coding of the attributes. What coding was used? Dummy coding or effects coding? Mention this in the text. It’s not clear at all.

13.What distributional assumptions did you make in your Hierarchical Bayes model for each attribute? Normal, lognormal, uniform? State this in text

14.What did the constant (alternative specific constant (asc)) represent in your utility function since your DCE adopted a forced choice format?

15.Tables 5 and 7: Could you clearly indicate which were the base levels (omitted categories/reference category) of the attributes? I can see you have used effects coding if I assume the base levels are first levels that appear for each attribute as the coefficient of the base level will be the negative sum of the included-category coefficients. However, you have not stated anywhere in the manuscript whether effects coding was used. You have to state this to make it easier for the reader.

16.In Tables 5 and &, provide the confidence intervals or standard errors for both the coefficients (relative utility) and standard deviations. There is only one confidence interval. Furthermore, what is the coefficient and standard deviation of your alternative specific constant (asc) and what does its coefficient represent?

17.Table 6. Which method was used to calculate the relative importance estimates (what you call “average importances”) probability analysis? Or Partial log likehood analysis? State this in the methods section and what it means.

18.The wording of the manuscript needs to be improved to reflect the fact that you are reporting a DCE.

19.Also place your study results in the context of Universal health coverage reforms in Kenya.

6. PLOS authors have the option to publish the peer review history of their article (what does this mean?). If published, this will include your full peer review and any attached files.

Reviewer #1: No

Reviewer #2: No

---

## [Author Response · Author response to Decision Letter 0]

16 Mar 2021

For the editors:

1. We have edited our manuscript to match PLOS One's style guidelines, including the naming of our files. 

2. We have removed supporting information files that were deemed non-essential for publication.

For reviewer 1:

1. All seven attributes in our study were identified from the literature. These attributes were confirmed as key components of a differentiated service delivery model with informants (researchers, healthcare workers) in Kenya who work with the HIV+ population. Of the original seven attributes that were identified, all were included in the final study. We did not identify any additional attributes that were later excluded. This explanation has been added to the text to clarify the process. 

2. Thank you for this comment. We opted for a partial profile design based on findings from Chrzan et al wherein respondents had difficulty cognitively processing choice tasks with more than six attributes. We have added an explanation for this design decision in the methods section. 

3. We addressed this very real concern by having the researcher present each choice task verbally and then allowing the patient to read it. This was done in a standardized way. For example, a researcher might present a choice task as follows: “If these care models were your only options, which one would you choose? The first option is a clinic with ART refills every 3 months, viral load samples at a health facility, and an individual support system. The second option is…” We have added an explanation for this process in the methods section.

4. I’ve added brackets to the units in Table 4 to improve clarity, as you suggested.

5. This is a great point. I have changed the wording in the methods section to reflect that the n=67 sample size calculation is for main effects. I also rephrased the limitation at the end of our discussion section as you suggested.

For reviewer 2:

1. I added a paragraph to the introduction section describing how HIV care has been managed over the past decade in Kenya. Later in the introduction, I also added a few sentences to explain why a DCE is the preferred method for this study.

2. In addressing the comments below, we hope to have clarified our methodology and provided important context on how the study was designed and how the data was analyzed. 

3. This paragraph was moved as you suggested.

4. It has now been clearly stated that our DCE uses a forced-choice format. Rationale for doing so has been included in the paragraph that you indicated.

5. Thank you for this comment. We opted for a partial profile design based on findings from Chrzan et al wherein respondents had difficulty cognitively processing choice tasks with more than six attributes. We have added an explanation for this design decision in the methods section. 

6. We used sawtooth software to generate the choice experiment design. Sawtooth generates the design by sampling from a subset of the full-choice designs for each respondent while ensuring level balance and near-orthogonality within each respondent’s profile, this allows for the generation of up to 999 blocks, and using a unique randomized design for each respondent reduces context effects (Reference Reed Johnson. Constructing Experimental Designs for Discrete-Choice Experiments: Report of the ISPOR Conjoint Analysis Experimental Design Good Research Practices Task Force). Sawtooth software is widely used for DCE design, administration and analysis, as a result we chose this approach for our experiment design. We have added these details to the manuscript.

7. This phrasing has been added to the text.

8. We addressed this concern by having the researcher present each choice task verbally and then allowing the patient to read it. This was done in a standardized way. For example, a researcher might present a choice task as follows: “If these care models were your only options, which one would you choose? The first option is a clinic with ART refills every 3 months, viral load samples at a health facility, and an individual support system. The second option is…” We have added an explanation for this process in the methods section.

9. We listed the exact tests used to summarize the sociodemographic variables. 

10. The wording has been changed as you suggested.

11. Per Orme et al, in standard probability analysis the average importance of an attribute is calculated by dividing its utility range (i.e. the utility of the highest level minus the utility of the lowest level) by the sum total lof utility ranges of all attributes. These calculations are done automatically by Sawtooth Software. This explanation has been added to the text.

12. The HB model in Sawtooth has two levels: At the upper level it is assumed that individuals’ vectors of part-worths are drawn from a multivariate normal distribution. At the lower level, a logit model is assumed for each individual, where the utility of each alternative is the sum of the part-worths of its attribute levels, and the respondent’s probability of choosing each alternative is equal to its utility divided by the sum of utilities for the alternatives in that choice set. Several Markov chain Monte Carlo (MCMC) simulations of an algorithm using these model estimates generates the part-worths for the individual, the mean for the population and variances and covariances. We have now included these details in the manuscript and added a reference to sawtooth HB analyses methods and formulae for readers to review.

13. This is addressed above in details regarding sawtooth analysis methods.

14. This is addressed above in details regarding sawtooth analysis methods.

15. We have now stated in the methods that effects coding was used.

16. Ninety-five percent confidence intervals have been added to Tables 5 and 7 for the standard deviations. The alternative specific constant has been addressed above in details regarding sawtooth analysis methods.

17. Average importances are calculated as indicated above, using estimated from HB analysis (Reference: Sawtooth software Interpreting Conjoint Analysis Data series: Interpreting Conjoint Analysis Data)

18. In addressing the comments above, we hope to have clarified our methodology and to now better reflect that we are reporting a DCE. 

19. We have added a paragraph in the discussion section to highlight how the fast-track, multi-month scripting model preferred by patients in our study may also help alleviate system inefficiencies, which would be beneficial for Kenya’s goal of providing universal healthcare to patients.

---

## [Decision Letter · Decision Letter 1]

20 Apr 2021

PONE-D-20-24745R1

Preferences of People Living with HIV for Differentiated Care Models in Kenya: A Discrete Choice Experiment

PLOS ONE

Dear Dr. Dommaraju,

Thank you for submitting your manuscript to PLOS ONE. After careful consideration, we feel that it has merit but does not fully meet PLOS ONE’s publication criteria as it currently stands. Therefore, we invite you to submit a revised version of the manuscript that addresses the points raised during the review process.

We look forward to receiving your revised manuscript.

Kind regards,

Matthew Quaife

Academic Editor

PLOS ONE

Journal Requirements:

Reviewers' comments:

Reviewer's Responses to Questions

**Comments to the Author**

1. If the authors have adequately addressed your comments raised in a previous round of review and you feel that this manuscript is now acceptable for publication, you may indicate that here to bypass the “Comments to the Author” section, enter your conflict of interest statement in the “Confidential to Editor” section, and submit your "Accept" recommendation.

Reviewer #1: All comments have been addressed

Reviewer #2: All comments have been addressed

2. Is the manuscript technically sound, and do the data support the conclusions?

Reviewer #1: Yes

Reviewer #2: Yes

3. Has the statistical analysis been performed appropriately and rigorously? 

Reviewer #1: Yes

Reviewer #2: Yes

4. Have the authors made all data underlying the findings in their manuscript fully available?

Reviewer #1: Yes

Reviewer #2: Yes

5. Is the manuscript presented in an intelligible fashion and written in standard English?

Reviewer #1: Yes

Reviewer #2: Yes

6. Review Comments to the Author

Reviewer #1: (No Response)

Reviewer #2: I am happy with the corrections. I will just request the authors to quickly write down the utility functions and insert it in the main text, after the paragraph explaining the HB model. Then submit the manuscript to the editor for publishing.

7. PLOS authors have the option to publish the peer review history of their article (what does this mean?). If published, this will include your full peer review and any attached files.

Reviewer #1: **Yes: **Jacob Kazungu

Reviewer #2: No

---

## [Author Response · Author response to Decision Letter 1]

28 Apr 2021

In addition to describing in words the functions for relative utility and attribute importance, we added the utility function used in our model which was derived from a textbook chapter on conjoint analysis by Rao et al. This is the utility function used by Sawtooth to calculate the relative utilities in our study. We hope that adding this function clarifies our approach and completes the revision of our methodology. 

The additional reference mentioned above has been added to our reference list.

---

## [Editor Report · Decision Letter 2]

9 May 2021

PONE-D-20-24745R2

Preferences of People Living with HIV for Differentiated Care Models in Kenya: A Discrete Choice Experiment

PLOS ONE

Dear Dr. Dommaraju,

Thank you for submitting your manuscript to PLOS ONE. After careful consideration, we feel that it has merit but does not fully meet PLOS ONE’s publication criteria as it currently stands. Therefore, we invite you to submit a revised version of the manuscript that addresses the points raised during the review process.

We look forward to receiving your revised manuscript.

Kind regards,

Basvarajaiah D. M., ph.D

Academic Editor

PLOS ONE

Journal Requirements:

Additional Editor Comments (if provided):

Dear author

I would like to acknowledge that I have reviewed your paper titled “Preferences of People Living with HIV for Differentiated Care Models in Kenya: A Discrete Choice Experiment.” It is a very important topic as countries aim for patient-centered care as well as health policy making decisions PLHIV (PONE-D-20-24745R2)

Below are a few comments that the authors can consider.

(i)The novelty of the research paper is very excellent; selection of the variables and attributes is patrsimonial state model. As per the literature, any model would be constructed or formulated by the author; the model should be expressive form and define the state variables of our objective of interest. I have carefully examined your model, you are unable to define the state variables in the model structure .Although, your formulated model will not be substantiate the state variables because not enough to propagate the state variable attributes for estimation of likelihoods based on numerical simulation. During the process of review, the following mathematical eqn affixed for your information and requested to include in your research paper

() = 11 + 22 + ⋯ + (1.1)

Ut_((xjt) ~) N (〖Dt〗_i 〖Ut〗_i )

The effect of each attributed of DCE was modeled by

Ut_xjt=Di((t1+t2..tn)/k)*Ui (1.2)

Where Di= The discrete value assigned for each of ith attributes

‘K’ is the discrete level of Xt

t1+t2..tn is the sum of the attributes at nth level

(ii) Table 7 represented the Normalized average utilities of all levels by gender. A higher magnitude indicates a stronger preference, while a positive or negative value .plz estimate the likelihood on each attributes based on the DCE state variables

iii) Table 3 Please correlate the selected attributes from weighting time for receiving ART drugs ( weighting time is the state variable)

The optimization of your model is reached maximum epoch, but interaction effect of the attributes is not mentioned in any ware either in result and discussion part.

Requested the Author,plz Rephrase the above limitation and comments .

---

## [Author Response · Author response to Decision Letter 2]

20 Jul 2021

Thank you for considering our manuscript for publication in your journal. Each comment from the editor in the third round of editing was helpful, especially in clarifying the methodology for our study. We addressed each comment and incorporated changes to our manuscript. Please find our responses below. Thank you.

Reviewer 1: 

1. The novelty of the research paper is very excellent; selection of the variables and attributes is patrsimonial state model. As per the literature, any model would be constructed or formulated by the author; the model should be expressive form and define the state variables of our objective of interest. I have carefully examined your model, you are unable to define the state variables in the model structure .Although, your formulated model will not be substantiate the state variables because not enough to propagate the state variable attributes for estimation of likelihoods based on numerical simulation. During the process of review, the following mathematical eqn affixed for your information and requested to include in your research paper

𝑈(𝑥𝑗𝑡) = 𝐷𝑡1𝑈𝑡1 + 𝐷𝑡2𝑈𝑡2 + ⋯ + 𝐷𝑡𝑟𝑡𝑈𝑡 (1.1)

Ut_((xjt) ~) N (〖Dt〗_i 〖Ut〗_i )

The effect of each attributed of DCE was modeled by 

Ut_xjt=Di((t1+t2..tn)/k)*Ui (1.2)

Where Di= The discrete value assigned for each of ith attributes

‘K’ is the discrete level of Xt

 t1+t2..tn is the sum of the attributes at nth level

The above equations defining our state variables have been added to our Methods section as requested.

2. Table 7 represented the Normalized average utilities of all levels by gender. A higher magnitude indicates a stronger preference, while a positive or negative value .plz estimate the likelihood on each attributes based on the DCE state variables

It is unclear what the reviewer is suggesting in this comment. If by “likelihood on each attribute” he is referring to the importance of each attribute by gender, there were no significant differences between genders when it comes to weight given to each attribute when choosing between care models. That is, the importance of each attribute for men and women were not significantly different from the importances of the cohort at large (listed in Table 6). As this was not relevant to our discussion, it was not included in our results section. I hope this clarifies the issue.

3. Table 3 Please correlate the selected attributes from weighting time for receiving ART drugs ( weighting time is the state variable)

Table 3 represents the sociodemographic information and baseline characteristics of our study population; there is no variable called “weighting time” in this table. I believe you are referring to Table 5, and specifically the attribute “frequency of ART refills.” If so, the weight of all attributes—and thus the corresponding relative utilities assigned to each level—are relative to the attribute with the highest importance. In our study, the most important attribute for patients when deciding between care models would be “location of clinical review,” not “frequency of ART refills.” I hope this clarifies your question.

4. The optimization of your model is reached maximum epoch, but interaction effect of the attributes is not mentioned in any ware either in result and discussion part.

When describing limitations of our study in our discussion section, we specifically mention that our study was not powered to examine two-way interactions between attributes. I have added an additional sentence to this paragraph to clarify, and I hope this addresses your comment.

---

## [Editor Report · Decision Letter 3]

22 Jul 2021

Preferences of People Living with HIV for Differentiated Care Models in Kenya: A Discrete Choice Experiment

PONE-D-20-24745R3

Dear Author 

We’re pleased to inform you that your manuscript has been judged scientifically suitable for publication and will be formally accepted for publication once it meets all outstanding technical requirements.

Kind regards,

D. M. Basavarajaiah, ph.D

Academic Editor

PLOS ONE
---

## [Editor Report · Acceptance letter]

9 Aug 2021

PONE-D-20-24745R3 

Preferences of People Living with HIV for Differentiated Care Models in Kenya: A Discrete Choice Experiment 

Dear Dr. Dommaraju:

I'm pleased to inform you that your manuscript has been deemed suitable for publication in PLOS ONE. Congratulations! Your manuscript is now with our production department. 

Kind regards, 

on behalf of

Dr. D. M. Basavarajaiah 

Academic Editor

PLOS ONE